# Brain-Gut and Microbiota-Gut-Brain Communication in Type-2 Diabetes Linked Alzheimer’s Disease

**DOI:** 10.3390/nu16152558

**Published:** 2024-08-03

**Authors:** Yomna S. Momen, Jayshree Mishra, Narendra Kumar

**Affiliations:** Department of Pharmaceutical Sciences, ILR College of Pharmacy, Texas A&M Health Science Center, Kingsville, TX 78363, USA

**Keywords:** gut-dysbiosis, gut-brain axis, prebiotics, probiotcs, transactional regulation, Janus kinase

## Abstract

The gastrointestinal (GI) tract, home to the largest microbial population in the human body, plays a crucial role in overall health through various mechanisms. Recent advancements in research have revealed the potential implications of gut-brain and vice-versa communication mediated by gut-microbiota and their microbial products in various diseases including type-2 diabetes and Alzheimer’s disease (AD). AD is the most common type of dementia where most of cases are sporadic with no clearly identiﬁed cause. However, multiple factors are implicated in the progression of sporadic AD which can be classiﬁed as non-modiﬁable (e.g., genetic) and modiﬁable (e.g. Type-2 diabetes, diet etc.). Present review focusses on key players particularly the modiﬁable factors such as Type-2 diabetes (T2D) and diet and their implications in microbiota-gut-brain (MGB) and brain-gut (BG) communication and cognitive functions of healthy brain and their dysfunction in Alzheimer’s Disease. Special emphasis has been given on elucidation of the mechanistic aspects of the impact of diet on gut-microbiota and the implications of some of the gut-microbial products in T2D and AD pathology. For example, mechanistically, HFD induces gut dysbiosis with driven metabolites that in turn cause loss of integrity of intestinal barrier with concomitant colonic and systemic chronic low-grade inﬂammation, associated with obesity and T2D. HFD-induced obesity and T2D parallel neuroinﬂammation, deposition of Amyloid β (Aβ), and ultimately cognitive impairment. The review also provides a new perspective of the impact of diet on brain-gut and microbiota-gut-brain communication in terms of transcription factors as a commonly spoken language that may facilitates the interaction between gut and brain of obese diabetic patients who are at a higher risk of developing cognitive impairment and AD. Other commonality such as tyrosine kinase expression and functions maintaining intestinal integrity on one hand and the phagocytic clarence by migratory microglial functions in brain are also discussed. Lastly, the characterization of the key players future research that might shed lights on novel potential pharmacological target to impede AD progression are also discussed.

## 1. Introduction

Global population growth is predicted to reach approximately 3.6 billion by the year 2100 [1]. This increase is mainly because of the developments in modern medicine, better hygienic practices, and the use of pesticides and antibiotics, which have raised life expectancy [2,3]. However, the worldwide increase in life expectancy led to a significant increase in the elderly population and age-related diseases. For instance, the senior population in Singapore and Malaysia is anticipated to rise by 372% and 277%, respectively, by the year 2030. In comparison, by the same year, it is expected that the aging population in France and the UK would have increased by 56% and 55%, respectively [4,5]. In the United States, the number of Americans aged 65 years and older is expected to increase from 58 million in 2022 to 82 million by 2050, and the government pays over 65% of healthcare spending on the elderly, with mean expenditure per person increasing from $12,411 in 2018 to $22,356 in 2020 [6,7,8].

Aging is a ubiquitous biological process characterized by a gradual, steady, and irreversible deterioration in cell and tissue functions across all organ systems, predisposing the individual to diseases such as diabetes, immune system diseases, Parkinson’s disease, depression, and dementia [9]. Dementia, although not a part of normal/typical aging, is a chronic progressive loss of cognitive function severe enough to impair daily life and independent function. Importantly, dementia is increasing progressively with an incidence of 55 million people worldwide [10]. Dementia has several forms, of which Alzheimer’s disease (AD) is the most common one, representing 60–80% of all dementia cases. Other types include vascular dementia, frontotemporal dementia, and dementia with Lewy bodies, associated with Parkinson’s disease [11,12,13]. By 2025, it is predicted that 13 million Americans aged 65 and above will have AD, marking a significant rise from the current estimate of approximately 6 million Americans in the same age group who are affected by AD. Furthermore, the estimated direct expenses for AD and dementia amount to $183 billion, with an estimated increase to $1.1 trillion by the year 2050 [14,15]. In other words, AD is the fastest-growing age-related disease that overburdens American society.

The primary neuropathological hallmarks of AD diagnosis are the extracellular deposition of the amyloid-β (Aβ) protein in the form of senile plaques and the intracellular accumulation of hyperphosphorylated microtubule-associated protein tau as flame-shaped neurofibrillary tangles [16]. The failure of the neurovascular unit to clear the Aβ from the brain parenchyma eventually results in its accumulation on cerebral blood vessels, a phenomenon known as cerebral amyloid angiopathy (CAA), which is the third hallmark of AD [17]. Aβ is produced upon two subsequent cleavages of the amyloid precursor protein (APP), a single-pass transmembrane protein predominantly expressed in CNS by β-secretase and γ-secretase. Early/familial-onset AD (EOAD/FAD) is associated with autosomal dominant mutations in genes encoding for APP as well as PSEN1/2 (presenilin 1 and 2), the catalytic subunit of γ-secretase, resulting in the overproduction of the insoluble toxic Aβ [18,19]. However, EOAD represents 1–6% of all AD cases, while most cases are sporadic, also called late-onset AD (LOAD), with an age of onset later than 65 years [20,21,22,23]. The clear cause of LOAD is still unknown but multifactorial contributors including age, environment, biology, and genetics can increase the risk for the disease. The apolipoprotein E ε4 (ApoE-ε4) allele is the most common genetic risk factor, followed by triggering receptor expressed on myeloid cell 2 (TREM2) mutations. ApoE-ε4 blocks Aβ uptake, decreases its clearance, and promotes the initial seeding of fibrillar Aβ deposition by competitive binding to Aβ receptors (such as low-density lipoprotein receptor-related protein1; LRP1) at the blood–brain barrier (BBB) and along the cerebrovascular system [24,25,26]. On the other hand, TREM2 mutations result in decreasing phagocytic activity of microglial cells. More recently, using genome-wide association studies (GWASs), novel genetic factors linked to a higher risk of LOAD have been identified including CD33, CR1, ABCA7, BIN1, CX3CR1, and SHIP1, which are also involved in microglial clearance of misfolded proteins and the inflammatory reaction [27,28]. Besides those genetic risk factors, environmental factors such as diet, sedentary lifestyle, and stress, exposure to environmental contaminants such as pesticides, detergents, heavy metals, and industrial byproducts, obesity, and type 2 diabetes (T2D) have been linked to the progression of sporadic AD. These factors significantly contribute to the pathogenesis of AD, possibly by affecting the gut microbiota diversity [28,29,30,31]. This review is focused on the current state of the literature on possible mechanisms by which the drivers of type-2 diabetes may promote changes in the gut microbiota, host molecular determinants such as transcription factors, and predisposition to AD through regulating the expression of the dialogue between gut and brain (and vice-versa) as a novel mediator as the possible mechanism for AD pathogenesis. First, we will discuss the levels of gut and brain communication followed by gut-microbiota-derived metabolites as a contributor to gut–brain communication and a key player in normal brain functions and AD pathology. 

## 2. Brain–Gut Axis

The communication between the gut and brain is bidirectional and complex, involving endocrine, immune, and neuronal communication mechanisms. The central nervous system (CNS) and the peripheral nervous system (PNS) are the two major divisions of the mammalian nervous system. While the CNS is made up of the brain and spinal cord, the PNS is made up of ganglia nerve branches that innervate various bodily organs [32]. Being a subdivision of the PNS, the autonomic nervous system (ANS) controls essential visceral functions in coordination with sympathetic and parasympathetic systems [33]. The vagus nerve is the main component that extends from the brainstem, innervating the gut and the enteric nervous system (ENS) and functioning as a superhighway carrying information from the gut to the brain and vice versa via vagal sensory afferents and motor efferents, respectively [34]. The ENS is the intrinsic nervous system of the gastrointestinal (GI) tract, containing approximately 400–600 million neurons, located within the walls of the GI tract, extending from the esophagus to the anal canal [34,35]. It is also known as the second brain of the body since it shares similarities with the brain in terms of structure, function, and chemical coding. It comprises two ganglionated plexuses: the myenteric plexus, which primarily controls the relaxation and contraction of the intestinal wall, and the submucosal plexus, which governs the functions and secretion of epithelial cells and regulates the flow of blood through the stomach. Being in the proximity of adaptive and innate immune cells, ENS neurons can regulate their functions and activities. Thus, the ENS serves as one of the regulators of the intestinal barrier by controlling multiple enteric functions, such as the immune response, nutrient detection, motility, microvascular circulation, and epithelial cells’ secretion of fluids, ions, and bioactive peptides [36]. It is evident that the vagal nerve and the ENS are in communication, with cholinergic activation via nicotinic receptors serving as the primary transmitter. This interaction allows the bidirectional flow of information between the gut and brain [37]. However, the ENS can also function independently of the vagus nerve as a result of its full reflex circuits, made up of sensory neurons and motor neurons; thereby, it regulates muscle activity and motility, fluid fluxes, and mucosal blood flow, as well as barrier function [32,34]. Besides the neuronal connection with the vagus nerve that physically links the gut and the brain, the hypothalamic–pituitary–adrenal axis (HPA) represents the hormonal route of gut–brain communication. On the other hand, enterochromaffin cells (ECCs) are interspersed in the gut epithelium and form close synaptic connections with certain vagal afferent fibers through cell extensions known as neuropods [38,39]. As primary sensors for intestinal nutrients, ECCs can release gut hormones and regulatory peptides postprandially such as cholecystokinin (CCK), serotonin, glucagon- like peptide-1 (GLP-1), Peptide YY (PYY), and gastric inhibitory polypeptide and via fasting such as ghrelin to modulate the HPA response to stress and nutritional/metabolic variations. Furthermore, these gut hormones and peptides can either reach the CNS via systemic circulation in an endocrine fashion or can act locally on the vagus nerve as neurotransmitters by targeting the chemoreceptors expressed on vagal sensory afferents, regulating short-term feelings of hunger and satiety as well as food intake and food choice (endocrine-mediated behaviors) to maintain energy homeostasis upon communicating the metabolic and nutrient status to the cognitive and emotional centers in the brain [34,39,40,41,42]. Besides the neuronal (vagus) and endocrine mechanisms of gut–brain communication, the gut immune system confers an additional pathway of the gut–brain communication network. The gut has both innate and adaptive immune cells. Dendritic cells of Peyer’s patches (lymphoid nodules embedded in the gut wall) and macrophages are crucial for innate immunity. Both are in close contact with the gut microbiota and maintain ENS homeostasis [43]. In addition, Mast cells are present in the mucosal (2–3% of lamina propria) and submucosal layers of the intestine and have a close anatomical relationship with the sensory autonomic nerve terminals. Mast cells have receptors for neurotransmitters and neuropeptides; therefore, their function is affected by nerve-derived substances. Other innate cells include natural killer cells and neutrophils. Besides that, the adaptive immunity represented by CD4^+^ and CD8^+^ T cells, as well as B cells, conveys signals to the enteric neural system, resulting in alterations to the ENS. It was reported that the vagus nerve can modulate the function of the immune cells in the gut, whereas, in the cholinergic anti-inflammatory reflex, the efferent nerve of the vagus nerve transmits the signal to the abdominal ganglia and then to the spleen via the β2 adrenergic receptor. Subsequently, it conveys the message to choline acetyltransferase^+^ T cells, which produce acetylcholine. The acetylcholine released by T cells acts on nicotinic acetylcholine receptors in macrophages, inhibiting the release of Tumor necrosis factor α (TNFα) [43,44]. 

Overall, the integration of neural (vagus) and hormonal (the HPA axis) communication pathways enables the brain to exert influence over the functions of various intestinal effector cells, including immune cells, epithelial cells, enteric neurons, smooth muscle cells, and enterochromaffin cells. On the other hand, these cells are under the influence of the gut microbiota, which, in turn, can modulate the immune, neural, and hormonal pathways of gut–brain communication. 

## 3. Gut

### 3.1. Gut Microbiota

The gastrointestinal tract is an intricately structured organ consisting of gut microbiota, the intestinal epithelium, and the mucosal immune system. Gut microbiota refers to the diverse ecosystem of microorganisms residing within the gastrointestinal tract that include bacteria, archaea, fungi, and viruses. Generally, the gut microbiota is composed of six phyla including *Firmicutes* and *Bacteroidetes*, which are considered the most predominant types, as well as *Actinobacteria*, *Proteobacteria*, *Fusobacteria*, and *Verrucomicrobia* [45]. Moreover, Candida, Saccharomyces, Malassezia, and Cladosporium are considered the most studied gut fungi [46]. Furthermore, while some bacterial species have been identified as possible pathogens, others are considered beneficial symbionts, coexisting in a mutually beneficial relationship with their human host. The imbalance in the ratio of these bacteria could make the host more susceptible to illness, for example, an increased or decreased *Firmicutes–Bacteroidetes* ratio has been associated with several pathological conditions such as obesity [47,48]. By converting dietary nutrients into microbial metabolites, the gut microbiota is not only able to communicate with each other and local gut cells but also communicate with organs that are distant from the gut such as the brain. Within the gut, microbial metabolites are often beneficial, boosting immune function and promoting tight junctions, but they can also be detrimental, triggering inflammation and the loss of intestinal barrier integrity [49,50,51]. Interestingly, either by passive or active transport, microbial metabolites can find their way out of the GI tract, entering circulation [52]. Once in circulation, these bacterial metabolites travel to different body locations, including the liver, kidneys, lymph nodes, reproductive tract, and even the brain [53]. Interestingly, gut microbiota has been linked to the regulation of immune and brain development and function in early life. For example, infants with high levels of *Bacteroides* showed improved cognitive outcomes, while those with high alpha diversity in their gut microbiota showed worse scores on the visual reception scale and expressive language scale. The colonization of gut microbiota during early life is pivotal for the development and maturation of the immune and endocrine systems, both of which affect CNS function [54]. Many studies using germ-free (GF) animals or broad-spectrum antibiotic-treated animals have shown the impact of the absence of gut microbiota on brain development, function, and behavior [55,56,57]. In that context, it has been reported that GF animals display memory dysfunction and altered cognitive functions attributed to the altered expression of the brain-derived neurotrophic factor (BDNF) in the hippocampus and cerebral cortex [58,59,60]. In addition, GF animals have also shown that the gut microbiota can affect stress reactivity and anxiety-like behavior by altering the expression of Gamma-aminobutyric acid (GABA) mRNA in the brain, inducing structural changes to neurons in the amygdala and causing variations in the degree of myelination in the prefrontal cortex [58,61]. Interestingly, the gut microbiota plays an important role in the regulation of microglial maturation and function, whereas significant microglial defects were observed in GF mice and antibiotic-treated mice along with a decrease in the number of immature phenotypes and altered inflammatory cytokine profiles that influence the basal surveillance (M0) state [62]. Therefore, the lack of gut microbiota in these animals provides convincing evidence for the involvement of the microbiota in brain development, physiology, and function.

### 3.2. Mechanistic Insights into Gut Microbiota and Brain Communication

The gut microbiota can directly affect the brain, primarily via its metabolites such as short-chain fatty acids (SCFAs), tryptophan metabolites, secondary bile acids, and trimethylamine N-oxide (TMAO), which can be transported to the brain via systemic circulation upon crossing the BBB. On the other hand, the gut microbiota can indirectly affect the brain by acting on the three routes of gut–brain communication: the neuronal, immune, and endocrine routes. In other words, the gut microbiota exerts effects on the vagus nerve, the ENS, the mucosal immune cells, and the enteroendocrine cells via its metabolites, locally acting neurotransmitters, and its cell wall components [28,40,44]. 

Short-chain fatty acids (SCFAs), such as acetate, propionate, and butyrate, are the main metabolites produced by the bacterial fermentation of dietary fibers [28]. The ability of the gut microbiota to generate enzymes for SCFA formation varies significantly. While the bacteria in the phylum *Firmicutes* are considered the predominant producers of butyrate, other genera of gut microbiota are thought to produce butyrate as well, including *Clostridium*, *Roseburia*, *Eubacterium*, *Anaerostipes*, and *Faecalibacterium* [63,64,65]. Interestingly, the communication between gut microbiota facilitates the production of SCFAs, whereas the acetate produced by *Bacteroides thetaiotaomicron* is further utilized by *Eubacterium hallii* to produce butyrate [66]. Other acetate-producing bacteria include *Bifdobacterium* spp. [67]. Furthermore, mucin-degrading bacteria such as *Akkermansia muciniphila* can produce both acetate and propionate upon mucin fermentation [68]. Once formed, SCFAs are absorbed by the colonocytes mainly via monocarboxylate transporters (MCTs), serving as an energy source for colonocytes [69,70]. In addition, SCFAs exert local effects in the gastrointestinal tract (GIT), whereas SCFAs facilitate the assembly of TJ proteins, enhancing the integrity of the intestinal epithelial barrier [70]. 

Interestingly, SCFAs can cross the BBB via MCTs, which are widely expressed in endothelial cells and brain tissue [71]. Once in the CNS, it can be recognized by neurons and other glial cells that consequently alter neurological and behavioral functions [72]. While butyrate administration may decrease microglia activation and lipopolysaccharide (LPS)-induced depression-like behavior in rats, SCFA treatment might cause functional alterations in microglia toward an anti-inflammatory and neuroprotective role [73]. Furthermore, it was found that SCFAs change the gene expression in astrocytes, whereas acetate upregulates the expression of the genes implicated in anti-inflammatory pathways while propionate enhances interleukin (IL)-22 expression in male cortical astrocytes, but not in females, demonstrating the sex-specific effects of SCFAs [74]. Moreover, treatment with sodium butyrate and the colonization of GF mice with an SCFA-producing bacterial strain have shown that SCFAs increased the expression of brain endothelial TJ proteins and decreased the permeability of BBB [75]. Additionally, SCFAs have been recognized as histone deacetylase (HDAC) inhibitors, thereby augmenting histone acetylation. This epigenetic alteration has the potential to amplify the activation of genes linked to synaptic plasticity and the sustenance of neurons, thereby inducing cognitive improvement in APP/PS1 and 5XFAD AD mice models [76,77,78,79]. In addition to HDAC inhibitory activity, other studies proposed that SCFAs could trigger the secretion of neurotrophic factors, including brain-derived neurotrophic factors, fostering persistence, expansion, and upkeep of neuronal cells [80]. Additional research revealed that butyrate treatment inhibited the activation of microglia M2 proinflammatory phenotype by suppressing NF-κB p65 phosphorylation, while nitric oxide formation was inhibited by acetate in primary rat microglial cells, thus exerting a neuroprotective effect [81,82]. Regarding its anti-neuroinflammatory effect, acetate was also reported to upregulate GPR41 and inhibit the ERK/JNK/NF-κB pathway in APP/PS1 mice [83]. Interestingly, SCFAs can indirectly affect the brain by binding to their G protein-coupled receptors (GPCRs) expressed on enteroendocrine cells, resulting in the secretion of GLP-1 and PYY, which can further regulate neuroinflammation. In addition, the reinforcement of the neuropeptide Y receptor by PYY was found to induce alterations in memory, social interaction, and sensorimotor function [84,85,86]. Overall, these indicate SCFAs as beneficial metabolites for normal brain function that protect against AD. 

Besides SCFAs, the gut microbiome can catabolize dietary tryptophan to active metabolites including indoles, kynurenine, and serotonin [87]. Indole and its directly related compounds, collectively called indoles, including indole-3-acetamine, tryptamine, indole-3-acetylaldehyde, and inole-3-pyruvate, are formed by members of *Proteobacteria*, *Firmicutes*, *Fusobacteria*, and *Bacteroidetes* that can produce tryptophanases [88,89,90]. Once dietary Tryptophan undergoes microbial conversion into indoles, it can travel through the lumen and gut mucosa into systemic circulation upon binding to the host’s aryl hydrocarbon receptors (AHRs), normally expressed on gut epithelial cells and CNS cells, including neurons, astrocytes, and microglia [91]. By acting on AHR, microbial-derived indoles are considered key regulators of immune homeostasis and epithelial integrity at barrier sites in the gut. It was reported that upon crossing the BBB, tryptophane metabolites can regulate microglial activation and its production of transforming growth factor alpha (TGFα) and vascular endothelial growth factor B (VEGF-B), thereby controlling the transcriptional program and the neurotoxic activity of astrocytes, limiting CNS inflammation [92]. Furthermore, indole can activate aryl hydrocarbon receptors in astrocytes, reducing inflammation via the regulation of the type 1 interferon [93]. In addition, indole derivatives, such as tryptamine, indole3-acetic acid, and indole-3-propionic acid, can regulate neuronal proliferation, differentiation, and survival through AHR signaling [87]. Furthermore, indole was found to promote the neurogenesis of adult brain tissue in mice, where neural growth is typically on the decline, upon systemic inoculation of indole or monocolonization with *E. coli*. This was attributed to the AHR-mediated signaling and the upregulation of β-catenin, Neurog2, and VEGF-alpha in the hippocampus [94]. Furthermore, indoles were found to prevent the pathological manifestations and neuroinflammation associated with AD via inhibition of NLRP3 (NOD-, LRR-, and pyrin domain-containing protein 3) inflammasome formation upon activation of the AHR, thereby improving cognitive function in an APP/PS1 mice model [95]. 

In addition, serotonin is a mood- and sleep-regulating neurotransmitter [90]. It is produced by epithelial enterochromaffin cells and, interestingly, gut microbiota, especially spore-forming bacteria, were found to be crucial for serotonin production [96]. Due to its polarity, gut-produced serotonin does not cross the BBB and it acts locally to improve the local integrity of the ENS, which then regulates gut motility and nutrient absorption [68,97]. Unlike serotonin, 5-hydroxytryptamine, which is an intermediate in the serotonin pathway, can cross the BBB and enter the brain where it is used for serotonin synthesis, where serotonin is also reported to be intrinsically involved in AD pathophysiology [98]. On the other hand, kynurenine is another tryptophane metabolite that is favorably produced under inflammatory conditions via the action of indoleamine 2,3-dioxygenase 1 (IDO1) and tryptophan 2,3-dioxygenase [39,99,100]. Kynurenine can cross the BBB and a higher level of kynurenine in the brain was correlated with depression-like behavior and the production of neuroinflammation and neurodegeneration, as seen in AD [101,102]. Furthermore, a deviation from mean kynurenine serum levels was correlated with neurological and dementia symptoms [103]. In addition, anthranilic acid and quinolinic acid are derivatives of kynurenine that have been reported to be neurotoxic [90]. Therefore, it can be concluded that different tryptophan metabolites have different effects on brain physiology and AD development.

Besides SCFAs and tryptophan metabolites, the gut microbiota can metabolize primary bile acids such as cholic acid and chenodeoxycholic acid synthesized by hepatocytes from cholesterol, conjugated to either taurine or glycine, and secreted with bile into the small intestine upon food intake into secondary bile acids [104]. Bile acids can signal either via cell membrane-bound or nuclear receptors, based on the lipophilicity properties and binding ability of the individual bile acid [105]. Therefore, secondary bile acids stimulate the transcription of the farnesoid X nuclear receptor (FXR) in the ileum and further trigger the synthesis of fibroblast growth factor 19, which, in turn, can enter the systemic circulation, cross the BBB, and eventually activate the arcuate nucleus of the hypothalamus [106]. Subsequently, the hypothalamus regulates glucose metabolism and suppresses HPA function [107,108]. Furthermore, the secondary bile acids can work on L cells in the ileum via the activation of Takeda G protein-coupled receptor 5 (TGR5). Upon activation of TGR5, it induces the release of GLP-1 from L cells regulating glucose homeostasis as well as uptake behavior and food intake [107,109]. Interestingly, secondary bile acids can also cross the BBB as lipophilic and activate TGR5 signaling in the brain and microglia. Then, TGR5 regulates GLP1 and PYY, which have a neuroprotective effect. While amyloid deposition and neurofibrillary tangles were reduced by GLP1, PYY increases phagocytosis and regulates inflammatory signaling [110,111]. Furthermore, additional research showed that the activation of TGR5 ameliorates Aβ- and LPS-induced cognitive impairment, NF-κB signaling, microglia activation, and neuronal apoptosis [112,113]. Additionally, chenodeoxycholic acid, a primary bile acid and potent agonist of FXR, showed neuroprotective effects against cognitive dysfunction, hippocampal APP processing, and insulin resistance in AD progression [114]. 

Moreover, TMAO is produced by the gut microbiome upon metabolism of choline-rich foods such as animal-derived products like red meat and full-fat dairy products. It can cross the BBB and has a direct effect on the CNS, accelerating brain aging and age-associated cognitive impairments and synaptic damage, suggesting that TMAO plays a role in AD pathology [115]. 

Besides microbial metabolites, the gut microbiota per se can produce catecholamines such as dopamine and norepinephrine and many other neurotransmitters such as serotonin and GABA or induce ECs to produce serotonin by its metabolites. However, neither catecholamines nor neurotransmitters can cross the BBB and they act locally in the gut or in the peripheral tissue, modulating the function of resident immune cells, which may further alter neurological function [33,44,86]. Despite that, it was found that GF mice have an increased turnover rate of dopamine, norepinephrine, and serotonin in the brain and the depleted levels of those neurotransmitters were restored by recolonization with spore-forming bacteria, but the mechanistic link between gut microbiota and the enhanced production of those neurotransmitters in the brain needs to be elucidated [56]. 

Hijacking vagus nerve signaling is considered the second fastest and most direct way for the gut microbiota to indirectly affect the brain [58]. The transmission of signals from the peripheral ends of the vagus nerve to the CNS, which innervate the muscle and the mucosa layers of the GI tract, is mediated either by the activation of mechanoreceptors that can sense the luminal volume or by the activation of chemoreceptors that can be triggered by chemical stimuli produced by enteroendocrine cells (EECs) such as hormones and neurotransmitters upon detecting the gut microbiota’s metabolites [116,117,118]. For example, serotonergic ECs, as one of the gut EECs, act as chemosensors for SCFAs, resulting in calcium signaling with the release of serotonin, which in turn relayed to specialized vagus nerve fibers that innervate the gut epithelium [118]. Furthermore, many animal studies have highlighted the role of gut microbiota–brain communication via the vagus nerve in modulating host behavior. For example, anxiety-like behavior in mice was modulated by *Lactobacillus rhamnosus* JB-1 due to the change in GABA receptor expression in the hippocampus and amygdala, as well as brain regions associated with fear and emotions. Interestingly, these findings were diminished in vagotomized mice, suggesting that the bacteria’s effects were mediated by neuronal communication to the brain [61]. Likewise, the vagus nerve was found to mediate the role of *Lactobacillus reuteri* in promoting social behavior in an animal model of autism spectrum disorder [119]. 

Besides gut microbiota’s effect on the vagus nerve, recent studies reported potential mechanisms of the gut microbiota–ENS interaction. The gut microbiome was reported to be partially involved in the development and function of the ENS [32]. In addition, as one of the major serotonin producers in humans, the gut microbiome can regulate the neuronal innervation of the colonic epithelium, the development of enteric glial cells via the release of serotonin, and the activation of its receptor in the ENS [120]. Moreover, the gut microbiome was reported to influence ENS activity and thereby gut motility in rodents via its cell wall components, SCFAs, and the metabolic products of bile acid [121,122,123]. Furthermore, GF mice showed increased activation of gut-extrinsic neurons connecting the brainstem sensory nuclei and gut sympathetic neurons, suggesting a suppressive effect of the gut microbiome on certain gut-to-brain signaling pathways [123]. Therefore, the interactions between luminal gut metabolites and the ENS and vagus nerve may be local in nature but with wide-reaching effects. Moreover, neuroactive molecules, such as GABA, secreted by several gut bacteria including *Bacteroides*, *Bifidobacterium*, and *Escherichia* spp., can modulate the synaptic transmission in the ENS, thereby influencing ENS homeostasis and disturbance such as acid secretion, gastric emptying, bowel motion, and the sensation of pain. On the other hand, GABA can eventually signal the brain function and behavior of the host by modulating neural signaling within the ENS [44,124,125].

To illustrate the connection between the HPA and gut microbiota, GF mice were used, where GF mice showed elevated plasma corticosterone indicating hyperresponsiveness to the HPA axis and the regulatory role of gut microbiota. Besides the aforementioned effect of gut microbiota-derived metabolites on EECs in HPA axis regulation, the gut microbiota can indirectly affect the HPA axis given the gut microbiota’s capacity to modify nutrient availability and the intimate connection between nutrient sensing and the secretion of peptides by enteroendocrine cells [40].

Collectively, these observations underscore the significant role played by gut microbiota on the different aspects of gut–brain communication, as well as the importance of examining the dysregulation of gut microbiota and, hence, its derived metabolites on AD pathogenesis. 

## 4. Compromised Gut–Brain Crosstalk in Alzheimer’s Disease

### 4.1. Gut Dysbiosis

Gut dysbiosis is the alteration of the composition of gut microbiota, marked by an excess of harmful or pathogenic microorganisms, a decline in beneficial bacteria, or an overall decrease in microbial diversity within the gut [126]. Gut dysbiosis is implicated in the development of cardiovascular diseases, diabetes mellitus, neuropsychiatric disorders, and neurodegenerative disorders [127]. Many factors are attributed to gut dysbiosis including genetics, the structure of the host’s intestinal wall, age, drugs including antibiotics, environmental factors, and, most importantly, diet. Both short-term [128] and long-term [129] dietary habits can drive alterations in microbial composition and diversity. The Western (American) diet (WD) is characterized by the excessive consumption of refined sugar, animal fats, eggs, processed meats (especially red meat), refined grains, salt, and potatoes with low intakes of fruits, vegetables, whole grains, and seeds; hence, the Western diet is low in fiber, vitamins, minerals, and other plant-derived molecules such as antioxidants [130,131,132]. Therefore, the major characteristic of the Western diet appears to be having high fat and low fiber content. Therefore, the next section will discuss the state of the literature on the effects of this fat consumption on the composition of the gut microbiota community. 

### 4.2. High-Fat-Diet-Induced Gut Dysbiosis and Leaky Gut Are Involved in AD Pathogenesis

High-fat diet (HFD) consumption leads to a profound change in the microbiota composition, primarily represented by the reduction in *Bacteroidetes* levels and an increase in *Proteobacteria* and *Firmicutes* levels in animals and humans (Figure 1). However, there is some inconsistency regarding microbiota changes driven by an HFD attributed to different types of fat that were used in various studies [86,133]. While *Bacteroidetes* are SCFAs-producing bacteria that increase TJ protein expression, maintain intestinal barrier integrity, and inhibit the growth of inflammatory bacteria, Firmicutes, e.g., *Proteobacteria,* are Gram-negative LPS-carrying proinflammatory bacteria that enhance proinflammatory cytokine production stimulation [132]. HFD-driven gut dysbiosis has been linked to the loss of intestinal barrier integrity, referred to as leaky gut. The intestinal barrier is a very complex system that is composed of gut microbiota, a mucus layer, an epithelial cell monolayer, immune cells in both lamina propria and submucosa layers, and proteins expressed on the luminal to the basolateral surface. The HFD-driven alterations in the gut microbiota with the consequent increased permeability of the gut barrier are mediated by various mechanisms. Firstly, the decreased expression of cystic fibrosis transmembrane conductance regulator and the Na-K-2Cl co-transporter 1 gene and protein reduce chloride secretion, resulting in alterations in the mucus phenotype and increased gut permeability. In addition, the alterations in the mucous layer are associated with the abundance of mucin-degrading bacteria. Furthermore, gut dysbiosis was also reported to affect the mucus layer composition and thickness by decreasing the number of goblet cells and lowering the mucus production, contributing to AD pathogenesis [134]. 

Besides the mucous layer, TJ proteins are crucial for intestinal barrier function. HFD has been implicated in decreasing *Bifidobacterium* spp. and *Lactobacillus* spp., gut microbiota that stimulate the gene expression of TJ proteins such as cingulin, OCLN, TJP1, and TJP2 in enterocytes. In addition, HFD causes a reduction in SCFAs-producing bacterial strains with the concomitant downregulation of TJ protein expression and the disassembly of TJ, which eventually causes the loss of the intestinal barrier. Consistent with that, a study has shown that the abundance of beneficial SCFAs-producing bacteria decreased intestinal inflammation by downregulating TLR4/NF-κB signaling and up-regulating tight-junction Claudin-2 in the colon, enhancing the intestinal barrier [135]. Furthermore, the stimulation of TLR4 by LPS can exacerbate the loss of intestinal barrier integrity. As an HFD causes the abundance of LPS-containing bacteria in the gut, the LPS is recognized by TLR-4 expressed on the intestinal epithelium, triggering a signaling cascade that results in the activation of a focal adhesion kinase (FAK). Subsequently, the activated FAK will, on one hand, increase the activation of myeloid differentiation primary response 88 (MyD88) and interleukin-1 receptor-associated kinase 4 (IRAK4), increasing intestinal permeability [136,137], and on the other hand, enhance the activation of NF-kB and mitogen-activated protein kinases (MAPKs), promoting inflammation [138,139]. Many studies have attributed the HFD-driven gut permeability to the increased levels of total secondary bile acids as deoxycholic acids with concomitant lower expression of genes of tight-junction proteins such as occludin [140,141,142]. One study showed that high deoxycholic acid in response to the HFD can downregulate AHR signaling, disrupting the differentiation function of ISCs, which in turn resulted in a decline in the number of goblet cells and MUC2 production [143]. In addition, AHR activation was found to maintain intestinal epithelial barrier function and mitigate colonic inflammation through the downregulation of IL-6, IL-7, and claudin-2 [144,145]. However, indoxyl sulfate, one of the tryptophane metabolites driven by gut dysbiosis, was found to induce intestinal epithelial cell damage by inducing reactive oxygen species (ROS) release and inhibiting mitophagic flux via interferon regulatory factor 1 (IRF1)-mediated suppression of dynamin-related protein 1 (DRP1) [146,147]. 

Increased gut permeability is accompanied by the uncontrolled translocation of luminal bacteria and harmful substances, such as LPS from pathogenic Gram-negative enterobacteria, into lamina propria, contributing to the initiation and progression of low-grade systemic inflammation, mediated by a TLR4-dependent proinflammatory response upon immune cell activation [86,148,149]. Interestingly, HFD-driven gut dysbiosis and leakiness are also accompanied by changes in BBB permeability, which in turn facilitates bacterial translocation into CNS, resulting in neuroinflammation, neuronal loss, neural injury, and ultimately, AD [75,150,151,152]. Interestingly, LPS was found near Aβ amyloid plaques in the hippocampus and neocortex brain lysates from AD patients [153,154]. LPS can activate the TLRs expressed on microglia, which are resident macrophages in the brain and are considered a key player in neuroinflammation [155]. Upon activation, microglia release a broad range of proinflammatory and toxic products such as reactive oxygen species, nitric oxide, and cytokines, resulting in neuroinflammation, and this is excessively engulfed through complement factors such as C1q and C3, resulting in neuronal loss and cognitive dysfunction [155,156]. However, the repeatedly activated microglia cells eventually lose their phagocytic effect, thus decreasing the degree of Aβ phagocytosis, inevitably developing plaque buildup.

Besides LPS-mediated neuroinflammation, the effects of gut dysbiosis on the brain are also mediated by bacterial metabolites that have detrimental functional consequences [157]. Many studies have shown a correlation between dysregulated bacterial metabolites and the progression of AD, involving microglia activation. For example, in a trial to identify novel biomarkers of dementia, a metabolomics study revealed that serum levels of acetate were found to be negatively related to the incidence of dementia, while a fecal metabolome analysis found that several SCFAs decreased with the onset and progression of AD and correlated with disease severity, including formic acid, acetic acid, propanoic acid, 2-methylbutyric acid, and isovaleric acid [158,159]. In addition, recent studies have documented alterations in SCFAs within AD models. These studies have observed elevated levels of butyrate, caproate, and 3-Methyvalerate in the serum of AD rat models [160]. Accordingly, compared to wild-type (WT) animals, AD transgenic mice had higher propionate levels and lower acetate levels. In addition, butyrate concentrations were higher in WT females in comparison with the other groups of mice, and altered fecal SCFAs were found to correlate with recognition memory and anxiety levels [161]. Another research study found that transgenic mice with elevated blood levels of acetate and propionate did not exhibit alterations in butyrate or lactate levels [162]. Additionally, research has demonstrated that the absence of bacteria that produce anti-inflammatory short-chain fatty acids (SCFAs) in the gut microbiota of AD patients, along with an increase in the inflammatory metabolite prostaglandin E2 (PGE2) in the brain, together enhance microglial activation and lead to chronic neuroinflammation [163]. On the other hand, compared to those without cognitive impairment, AD patients have higher levels of TMAO in their cerebrospinal fluid (CSF), which has also been connected to Aβ aggregation and neuronal degeneration [164,165]. It has also been discovered that rodents’ cognitive impairment is partly caused by elevated circulating TMAO levels [166,167]. Since TMAO downregulated the hippocampal antioxidant enzyme methionine sulfoxide reductase, this was linked to increased microglia activation, brain inflammation, and the production of reactive oxygen species (ROS) in the hippocampus [166]. Furthermore, an interesting study revealed that intestinal dysbiosis and the development of AD are highly common in individuals with AD, with a large perturbation of tryptophan metabolism, particularly indole derivatives and serotonin production [158]. Furthermore, indole derivatives demonstrate diverse patterns of expression, with both upregulation and downregulation depending on the derivative. While indole-3-propionic acid and indole-3-pyruvic acid showed high levels in AD and were considered predictive factors of AD progression, 5-hydroxyindole, indole-2-carboxylic acid, and 3-(2-hydroxyethyl) indole showed reduced levels [158,168]. The later indole derivatives have an inhibitory effect on the microglia’s response to inflammation, as previously mentioned. In contrast, the serum level of indoxyl sulfate significantly increases in many neuropsychiatric disorders, dementia, and AD [169]. Additionally, several studies reported elevated CSF levels of kynurenic acid, quinolinic acid, and anthranilic acid in AD patients, while others showed elevated brain kynurenic acid levels mainly in the hippocampus of AD patients [103,170,171]. Furthermore, a previous study has shown that WD-driven gut dysbiosis dysregulated bile acid synthesis by lowering the gene expression of the enzymes involved in the synthesis, which eventually compromised TGR5 signaling in the brain and microglia, resulting in systemic inflammation, microglial activation, and reduced neuroplasticity with ultimate cognitive impairment [172]. It is worth mentioning that TGR5 regulates GLP1 and PYY, which have a neuroprotective effect. GLP1 reduces amyloid deposition and neurofibrillary tangles while PYY increases phagocytosis and regulates inflammatory signaling [110,111]. Moreover, it was discovered that Aβ-triggered neuronal apoptosis in differentiated SH-SY5Y cells and mouse hippocampal neurons was accompanied by increased FXR mRNA and protein expression. This, in turn, exacerbated Aβ-triggered neuronal apoptosis by weakening cAMP response element binding protein/brain-derived neurotrophic factor (CREB/BDNF) signaling [173]. A study found a significant correlation between worse cognition and increases in the ratios of secondary to primary BAs by gut microbial enzymes as well as numerous conjugated BAs [174]. Furthermore, it was discovered that in neuronal C/EBPβ transgenic mice, Bacteroides fragilis and their metabolites 12-hydroxy-heptadecatrienoic acid (12-HHTrE) and PGE2 activate microglia and cause AD pathology [175]. 

In summary, most of the studies linking gut dysbiosis and AD have indicated that dysregulated bacterial metabolites were correlated with the dysregulated expression of genes that either regulate the microglia phagocytosis and inflammatory response or maintain the integrity of the intestinal barrier and the BBB. However, the mechanism by which these metabolites regulate gene expression remains elusive. 

## 5. Diet and Transcription Factors in AD Progression

Increased gut permeability and systemic low-grade inflammation contribute to high-fat diet (HFD)-related disorders. Recent findings have shown that colonic inflammation precedes other tissues and ultimately results in distal organ inflammation such as in the brain, predisposing individuals to AD (Figure 2). Many of the effects of an HFD in the stem and progenitor cell compartment are driven by proliferator-activated receptor-δ (PPAR-δ), which augments ISC self-renewal and bestows features of stemness on non-stem cell progenitors, resulting in colorectal cancer initiation and progression [176,177]. HFD- induced obesity and insulin resistance through the upregulation of PPARγ and AMPK downregulation in liver and adipose tissue, an effect that was reversed by SCFAs [178]. Additionally, HFDs with Enterobacter cloacae B29, an opportunistic pathogen, were found to enrich PPAR signaling and upregulate its downstream target genes, involved in lipid and lipoprotein metabolism in the colons of obese GF mice [179]. AHR is a ligand-activated transcription factor that contributes to development, the immune response, and metabolite signaling, and suggests a significant role in endogenous metabolism in addition to its role in sensing xenobiotics and controlling the expression of genes involved in xenobiotic metabolism, such as drug transporters and drug-metabolizing enzymes [180]. Interestingly, AHR signaling in hepatocytes can be stimulated by ox-LDL, one of the WD’s components. ox-LDL can activate TLR2/4 signaling cascades that ultimately provide high levels of activated IDO1, which in turn produce excess kynurenine from the available tryptophan, thereby upregulating AHR signaling in hepatocytes and leading to obesity and associated diseases [181]. Furthermore, it was shown that the inhibition of AHR protects against and cures obesity possibly by lowering the expression of the PPARα-target genes, Cyp1b1, Scd1, and Spp1 in the liver [182]. 

Interferon regulatory factors (IRFs) are a group of transcription factors that are activated downstream of TLRs, and upon activation, they dimerize, translocate into the nucleus, and bind to IFN-stimulated regulatory elements (ISREs) in DNA, starting the expression of interferons (IFNs). Although their predominant function is in innate immune responses and oncogenesis, IRF family members are emerging as metabolic transcriptional regulators in obesity/type-2 diabetes (T2D) as well. IFN induces gene expression via the Janus kinase (JAK)-signal transducer and activator of transcription (STAT) pathway (JAK1, 2, TYK2) (STAT1, STAT3, STAT4, and STAT5A/B) resulting in the expression of a large spectrum of ISGs [183,184]. IRF3, 5, and 7 play roles in HFD-induced obesity and insulin resistance by mediating systemic chronic low-grade inflammation by increasing proinflammatory gene expression in adipocytes, resident and infiltered macrophages in adipose tissue, liver tissue, and skeletal muscle [185,186,187,188,189]. Likewise, STATs are a family of seven members including STATs 1–4, 5A, 5B, and 6, and the major upstream kinases required for STAT activity are JAK proteins. Adiposity, energy expenditure, glucose tolerance, and insulin sensitivity are only a few of the metabolic processes that are influenced by JAK/STAT signaling in peripheral metabolic organs. There is growing evidence that the JAK/STAT signaling pathway is dysregulated in some metabolic diseases such as obesity and T2D. On a molecular level, obesity is characterized by high levels of IL-6 in white adipose tissue that, in turn, chronically activate intracellular JAK-STAT3 signaling and increase the expression of suppressor of cytokine signaling-3 (SOCS-3), which negatively regulates IL-6 signaling, hinders insulin action, and eventually results in obesity and insulin resistance. In addition, the JAK-STAT1 pathway plays a key role in the ability of IFN-γ to induce insulin resistance, decline triglyceride stores, and down-regulate the expression of lipogenic genes in mature human adipocytes. The increased IFN-γ levels and JAK-STAT1 signaling in obesity contribute to adipose tissue dysfunction and insulin resistance [190]. Furthermore, one study revealed that *Cldn10* and *EGF* have lower expression in the intestinal tissues of HFD-fed mice, impairing intestinal barrier function [191]. In addition, indoxyl sulfate inhibited the nuclear factor erythroid 2 (NFE2)-related factor 2 (Nrf2) nuclear translocation as well as its downstream-related antioxidant enzymes, resulting in a loss of intactness of intestinal epithelial cells due to oxidative damage [147]. Moreover, a transcription factor (TF) interaction analysis and DEG-microRNAs (miRNAs) interaction analysis revealed that the transcriptional regulators for the differentially expressed genes common in tissues of patients with T2D and neurological diseases are FOXC1, GATA2, FOXL1, YY1, E2F1, NFIC, NFYA, USF2, HINFP, MEF2A, SRF, NFKB1, USF2, HINFP, MEF2A, PDE4D, CREB1, SP1, HOXA5, SREBF1, TFAP2A, STAT3, POU2F2, TP53, PPARγ, and JUN, suggesting a link between T2D and AD [192]. The upregulation of NF-κB and STATs along with the downregulation of nuclear factor erythroid 2 (NFE2)-related factor 2 (Nrf2) and Peroxisome proliferator-activated receptor-gamma coactivator-1alpha (PGC-1α) mediate HFD-induced neuroinflammation and obesity-induced oxidative stress [193,194,195,196,197]. Furthermore, inflammation, associated with HFD-mediated obesity and diabetes, activates the transcriptional factor CCAAT-enhancer-binding protein (C/EBPβ), which activates the transcription of the cysteine protease asparagine endopeptidase (AEP). AEP enhances the cleavage of both APP and Tau, promoting their deposition, and eventually AD pathology [198,199]. A palmitate-rich diet was found to induce sterol regulatory element binding protein-1 (SREBP1) expression and activation, underlying the increased BACE 1 activity and Aβ genesis [200]. It was reported that NF-κB, specificity protein 1(Sp1), Yin Yang 1 (YY1), hypoxia-inducible factor 1α (HIF-1α), and PPARγ transcriptionally regulate BACE1 expression, and hence APP processing and Aβ generation [201,202]. However, further studies are needed to investigate whether HFD can upregulate the transcriptional activity of YY1 and SP1, predisposing individuals to obesity-related cognitive deficits. Similarly, the effect of an HFD on the Erythroblast Transformation-Specific (Ets) family of transcription factors, which mediate cellular responses to environmental stimuli downstream of MAPKs, remains to be elucidated. Interestingly, Ets transcription factors ER81 and Elk1 were found to bind to the proximal promoter elements of the human PSEN1 gene and activate PSEN1 transcription, which in turn, according to the amyloid hypothesis, leads to the increased production of Aβ42 and deposition of amyloid plaques in the brains of FAD [203,204]. Moreover, the retrograde endocannabinoid signaling cascade and the GABAergic synapse pathway—two important pathways implicated in the pathophysiology of AD—are regulated by ELK-1, GATA1, and GATA2. [205,206]. Furthermore, significant regulatory TFs (FOXC1, GATA2, YY1, FOXL1, NFIC, E2F1, USF2, SRF, PPARγ, and JUN) were found to regulate the function of the differentially expressed genes in AD [207]. On one hand, E2F1 was reported to play a role in metabolism whereas it enhances adipogenesis, inhibits lipolysis and β-oxidation in adipose tissue, enhances lipogenesis, and inhibits cholesterol transport in the liver. It is not surprising that E2f1 mRNA and protein levels increased in the visceral white adipose tissue of obese human subjects and are positively correlated with insulin resistance and circulating free fatty acids [208]. In addition, E2F1 expression was also increased in the visceral adipose tissue of mice fed on HFD and in leptin-deficient (ob/ob) mice, two widely used mouse models of obesity [209]. Furthermore, liver biopsy samples from diabetic individuals had higher expression of E2F1 mRNA, which is correlated with PCK1 levels and may be related to the development of hyperglycemia in humans as well as mice [210]. On the other hand, E2F1 was upregulated in AD brains, mediating abortive cell cycle progression and promoting apoptotic neuronal cell death [211,212]. Interestingly, the components of the HFD have been reported to induce metabolic reprogramming of microglia cells, the central pathophysiological player in AD pathogenesis. In the early stages of AD, microglia engage in protective functions such as Aβ clearance, the suppression of tau phosphorylation, and the release of neurotrophic factors, primed by the M1-like microglia response and maintained by M2 microglia. However, as the disease progresses, persistently activated microglia contribute to neurodegeneration by inhibiting the uptake of Aβ and the degradation of internalized Aβ, as well as triggering tau hyperphosphorylation due to the predominance of the proinflammatory M1 phenotype [213,214,215]. This disease-associated microglia phenotype is accompanied by metabolic alterations whereas microglia can shift to a preferential reliance on glycolysis with dysfunctional mitochondrial respiration, known as metabolic reprogramming [216,217]. It is now becoming increasingly appreciated that microglia’s metabolic reprogramming is dependent on the available nutrients and environment and determines their phenotype and function [218]. Nutrients such as fructose and cholesterol were found to mediate the metabolic reprogramming of microglia, driving disease-associated microglia with the concomitant release of inflammatory cytokines in AD brains [219]. Furthermore, saturated fatty acids have been shown to act as ligands for TLRs, promoting pro-inflammatory gene expression in BV2 and primary mouse microglia via the activation of transcription factor NF-κB [220]. HFDs downregulate the transcriptional activity of PPARγ in microglia, resulting in a disease-associated proinflammatory phenotype of microglia [221]. Moreover, a study investigating the transcriptomic changes in AD-associated astrocytes and microglia in the brains of mice fed HFDs and humans with AD revealed that three shared genes were upregulated in both HFDs and AD including *C4b*, *Kcnj2*, and *Ddr*1, which mediate synaptic pruning, microglia proliferation, migration, and the inflammatory response, suggesting a mechanism through which HFDs cause AD [222].

Apart from this, a maneuverable actin cytoskeleton is crucial to maintaining the surveillant, migratory, and phagocytic functions of microglia in response to chemotactic factors, whereas the coordinated polymerization of actin filaments provides a protrusive force (lamellipodia) and a thin filamentous protrusion to sense and direct the migration (filopodia) [223]. To form lamellipodia and filopodia, the G-actin monomers are arranged into oligomers, a step known as nucleation. Following nucleation, oligomers continue to polymerize to create F-actin, a process that is reliant on the amount of G-actin that is accessible in the cytoplasm. F-actin polymers assemble to create intricate networks with branches. Branches in the actin network must develop for lamellipodia to form and are essential to direct motility. Eventually, F-actin undergoes disassembly and depolymerization via actin-severing proteins such as gelsolin, promoting the reorganization of the actin cytoskeleton and maintaining functional microglial cells [224]. Even though the disruption of actin dynamics does not affect the transcription of genes involved in the classical microglia inflammatory response, it resulted in decreased NO secretion and reduced release of TNFα, IL-6, and IGF-1, with the loss of their neuroprotective capacity and the dampening of microglia migration, proliferation, and more importantly, phagocytosis [225]. In addition, gelsolin was found to reduce amyloid-ß levels in APP/PS1 transgenic mice, suggesting that it plays a protective role in Alzheimer’s disease [226]. On the other hand, alterations in microglial motility-related proteins were observed during aging, which were more pronounced in the presence of AD pathology [224].

Gelsolin is an eponym for a conserved class of actin-binding proteins (ABPs), where all members contain the homologous gelsolin-like (G) domain. Besides gelsolin, the family includes villin, adseverin, capG, advillin, supervillin, and flightless I, which are all transcriptionally regulated by SRF [227,228]. During F-actin polymerization, the increased Rho-GTPase activity of SRF depletes the nucleus G-actin pool. Cofactors like MRTF (myocardin-related transcription factor) can alter the activity of SRF. Context-dependent gene transcription is activated by MRTF binding to SRF in the nucleus, which is triggered by depleted G-actin pools [229]. Therefore, whether SRF links obesity/diabetes and AD in terms of the remodeling of the microglia actin cytoskeleton needs to be elucidated.

## 6. Role of Kinases in AD

The Janus kinase (JAK) family consists of non-receptor tyrosine kinases JAK1, JAK2, JAK3, and Tyrosine-kinase 2 (TYK2). While JAK1, 2, and TYK2 are ubiquitously expressed, JAK3 was thought to be expressed in the bone marrow and lymphatic system, as well as endothelial cells and vascular smooth muscle cells [230,231,232,233,234,235]. JAKs mediate the signal initiated by cytokines and growth factors upon binding to their receptors. Once the cytokine binds to its receptor, two JAKs are recruited to the receptor and induce their auto- and transphosphorylation. JAKs phosphorylate the intracellular tail of the cytokine receptor on specific tyrosine residues, which in turn act as docking sites for two STAT proteins, forming a dimer. The dimer enters the nucleus, starting gene expression upon binding to DNA and promoting cell proliferation and survival signal [236]. JAKs have seven homology domains (the JAK homology domain, JH). The first JH, referred to as the kinase domain, JH1, has a length of around 250 amino acid residues, begins at the carboxyl terminus, and can phosphorylate substrates. Although it lacks kinase activity, JH2 is a Pseudo Kinase domain that shares structural similarities with JH1. Its primary function is thought to be controlling the kinase activity of the JH1 domain. While JH3 with one-half of JH4 constitutes the Src-homology 2 (SH2) domain, the combination of one-half of JH4, JH5, JH6, and JH7 constitutes the FERM domain. The primary regulators of JAK and cytokine-receptor membrane-proximal box1/2 region binding are SH2 and FERM domains [237]. The FERM Domain also mediates the interaction with the Cytokine receptor and interacts with the JH1 domain, enhancing its kinase activity, while SH2 has a negative regulatory effect on the FERM domain. In that context, the intramolecular interaction between the FERM and SH2 domains of non-phosphorylated JAK3 prevents it from binding to villin, one of the actin-severing proteins, while the autophosphorylation of JAK3 at the SH2 domain releases its hindrance over the FERM domain, allowing it to bind villin [238]. 

Much of JAK3’s role has also been reported in several functions of the epithelial mucosa including intestinal mucosal wound healing and homeostasis, which is a stepwise process that proceeds along a continuum starting from the epithelial–mesenchymal transition (EMT), cell migration, proliferation, and the termination of EMT and ending with cell differentiation. JAK3 contributes to each step. Mechanistically, intestinal epithelium-specific cytoskeletal protein villin is phosphorylated by JAK3, which is activated by IL-2 produced from immune cells underneath the mucosal epithelial layer. Next, JAK3-phosphorylated villin actin-severing activity facilitates actin reorganization via the JAK3-villin complex. The actin polymerization and bundling near the wound edge are facilitated by the new actin nuclei produced by these severing activities. In addition, JAK3 phosphorylates p52ShcA (Shc), which in turn stimulates Ras activation to enhance cell growth and proliferation through the promotion of intestinal epithelial cell proliferation. Additionally, JAK3 mediates EMT by reinforcing the adherens junction (AJ), which is required following the successful completion of the proliferative phase. Mechanistically, JAK3 maintains β-catenin localization at the AJ upon phosphorylation at Tyr30, Tyr64, and Tyr86 [239].

In terms of JAK3’s neurological function, it was shown that while JAK3 inhibition can boost neuronal differentiation with lengthy neurite outgrowth and their maintenance, JAK3 is necessary for controlling neurite development and microglial differentiation in the spinal cord [240]. Furthermore, the upregulation of JAK3/STAT6 enhanced microglia M2 polarization, alleviating M1-mediated demyelination and neuroinflammation in an encephalomyelitis (EAE) mouse model, and improved myelin recovery and ameliorated the symptoms of inflammatory demyelinating diseases, such as multiple sclerosis [241,242]. In addition, JAK3 was found to enhance the expression of the Ca(2+)-activated K (K+) channel, KCa3.1, increasing the migratory capacity of alternatively activated microglia [243].

## 7. Concluding Remarks and Future Work

In this review, we discussed the current state of the literature based on our understanding of the essential roles and levels of gut–brain communication and the contribution of gut microbiota-derived metabolites in such crosstalk. We further delineated the neural–hormonal communication pathways enabling the brain to exert influence over various intestinal effector cells and how these cells are under the constant influence of the gut microbiota, which, in turn, modulate the immune, neural, and hormonal pathways of gut–brain communication. Furthering these, we delved into how the lack of gut microbiota provides evidence for its involvement in brain development, physiology, and function. This was in part because the gut microbiota provides various amino-acid derivatives, for example, those of tryptophan metabolites, which have different effects on brain physiology and AD development. These underscore the significant role of gut microbiota on the different aspects of gut–brain and vice-versa communication and the impact of an imbalance in gut microbial communities and their derived metabolites on T2D and the link to AD pathogenesis. Since diet shapes gut-microbial communities, this article went on to characterize the impact of the Western diet, which is rich in high-fat and low-fiber content, on the composition of the gut microbiota community and the imbalance in metabolite derivates. At the end, we summarized studies linking imbalanced metabolites in T2D and AD and speculated that this could be related to the dysregulated gene expression that may either regulate the microglia phagocytosis and inflammatory response or maintain the integrity of the intestinal barrier and BBB. However, the mechanism by which these metabolites regulate gene expression remains elusive. Therefore, we foresee future research on the mechanistic aspects of such gene expressions with a future goal of using this information to develop pharmacological, probiotic, and prebiotic classes of interventions that can not only restore the impact on genes but also educate people on how to take preventative or amelioration measures for diabetes that can prevent or at least significantly slow down the progression of the symptoms of dementia as seen in Alzheimer’s disease and other neuroinflammatory conditions.

## Figures and Tables

**Figure 1 nutrients-16-02558-f001:**
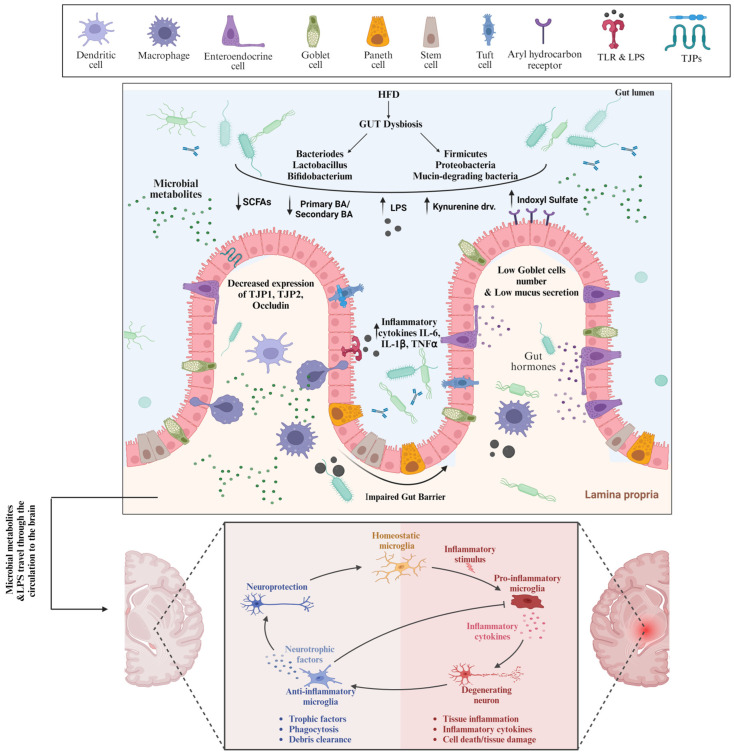
High fat diet-induced Gut Dysbiosis and Leaky Gut are involved in AD Pathogenesis. Consumption of high fat diet causes gut dysbiosis in terms of predominance of pathogenic bacteria and the harmful metabolites which bind to receptors on the surface of intestinal epithelial cells and other cells, inducing deleterious cellular responses and ultimately loss of intestinal barrier integrity. As a result of leaky gut, the pathogenic bacteria and metabolites travel through the systemic circulation to the brain where they induce the microglia cells polarization, and eventually neuroinflammation. TLR: Toll like receptors, LPS: Lipopolysaccharide, TJPs: tight junction proteins, SCFAs: short chain fatty acids, IL-6: Interleukin-6, TNFα: Tumor necrosis factor-α.

**Figure 2 nutrients-16-02558-f002:**
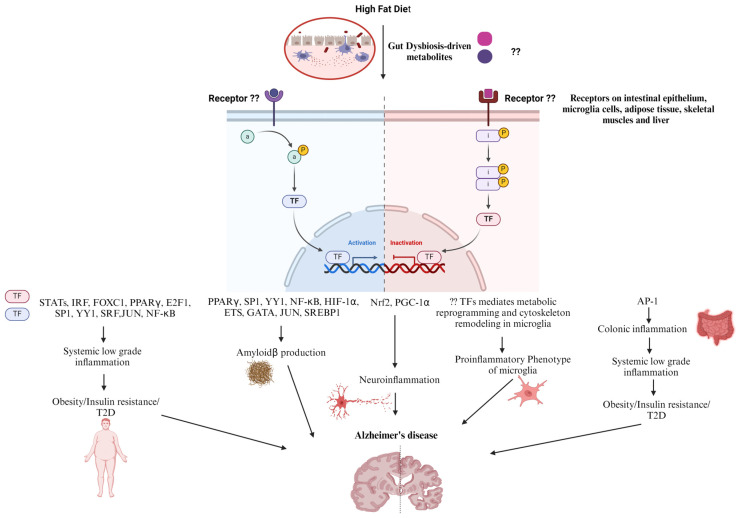
Perspective on HFD-dysregulated Transcription Factors Contribution in AD Progression. High fat diet induces gut dysbiosis. Upon binding to cell surface receptors, the gut dysbiosis-driven metabolites dysregulate downstream transcription factors. Those transcription factors regulate the expression of genes implicated in colonic inflammation, insulin resistance, type 2 diabetes, and eventually systemic low-grade inflammation that indirectly leads to Alzheimer’s disease progression. On the other hand, genes related to microglial cells metabolic reprogramming and cytoskeleton remodeling, as well as Amyloidβ production are dysregulated resulting in neuroinflammation ending with Alzheimer’s disease.

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
