# Peer review of "High Fat Diet-Induced Dysregulation of Tyrosine Kinases Is a Novel Player in Gut–Brain Axis in Alzheimer’s Disease"

_nutrients, 2024, doi:10.3390/nu16152558_

Round 1

Reviewer 1 Report

Comments and Suggestions for Authors

The authors try to summarize and analyze the brain-gut and microbiota-gut-brain communication in Type-2 diabetic linked AD. This is an interesting topic to work on. However, there are some concerns on the article:

1, The authors only generally discuss the brain-gut and microbiota-gut-brain communications. They have not discussed on diabetic linked AD specifically. The content and the title are not relevant. 

2, the recent advance in GLP-1 agonist against neurodegeneration in PD and AD has been achieved. The authors can elaborate more on this aspect.

3, some sections such as subtitle 6 Role of kinases in AD and 5 Diet and Transcription Factors in AD Progression are not related to brain-gut and microbiota-gut-brain communication. The authors should focus on their relevant topics 

Author Response

Comment 1:-The authors only generally discuss the brain-gut and microbiota-gut-brain communications. They have not discussed on diabetic linked AD specifically. The content and the title are not relevant.

Response 1: I agree with you that the title is not relevant to the topic of the review article. This review article provides a new perspective of gut-brain communication in terms of transcription factors as a commonly spoken language facilitates the interaction between gut and brain of obese diabetic patients who eventually develop cognitive impairment. That commonality is in terms of regulation of JAK3 expression both in gut and brain and hence maintaining the integrity of intestinal barrier and the phagocytic as well as migratory microglial functions, probably by regulation of microglial actin cytoskeleton remodeling, thereby alleviating the systemic chronic low-grade inflammation as well as Aβ deposition. Moreover, the characterization of those common transcription factors might represent a novel potential pharmacological target to impede AD progression.  

Therefore, I suggest to change the title to "High Fat Diet-induced Dysregulation of Tyrosine kinases is a Novel Player in Gut-Brain Axis in Alzheimer’s Disease". and please find the changes in the abstract. 

Comment 2: the recent advance in GLP-1 agonist against neurodegeneration in PD and AD has been achieved. The authors can elaborate more on this aspect.

Response 2: Thank you for your comment. However, the review article is not discussing GLP-1 agonist in neurodegeneration. It is not related to the review topic or idea.

Comment 3:some sections such as subtitle 6 Role of kinases in AD and 5 Diet and Transcription Factors in AD Progression are not related to brain-gut and microbiota-gut-brain communication. The authors should focus on their relevant topics.

Response 3: The main idea of this review article is to show how the high fat diet (via gut dysbiosis and driven metabolites) may affect the transcription of tyrosine kinases, especially JAK3 in both gut and brain in obese diabetic patients, resulting in loss of the integrity of intestinal barrier and the phagocytic as well as migratory microglial functions, probably by regulation of microglial actin cytoskeleton remodeling, thereby eventually causing the systemic chronic low-grade inflammation ,  Aβ deposition and cognitive impairment commonly seen in obese diabetic patient with Alzheimer's disease. Therefore, discussing the role of Tyrosine kinases is relevant to the review topic. Please find the clarifying sentence added at the end of subtitle 6.

Reviewer 2 Report

Comments and Suggestions for Authors

In the article, the authors discuss the role of gut microbiota in the development and progression of Alzheimer's Disease (AD) in the context of Type-2 Diabetes (T2D). They focus on specific mechanisms by which gut dysbiosis might influence neuroinflammation and cognitive decline. The topic of the article is highly relevant and timely, given the increasing interest in the interplay between gut microbiota, metabolic disorders, and neurodegenerative diseases. The novel aspect of linking type-2 diabetes (T2D) with Alzheimer’s disease (AD) through microbiota-gut-brain communication pathways provides a fresh perspective that is not widely covered in the current literature. However,  there are some areas for improvement:

Including more discussions on conflicting results and different perspectives in the literature could provide a more balanced view. This would also help in highlighting the areas where further research is needed. Could you include a more detailed discussion of these results and the alternative viewpoints proposed in the literature in sections Gut microbiota-brain Axis or Diet and Transcription Factors in AD Progression sections?

While the review touches on potential interventions, it could be improved by offering more detailed discussions on the clinical implications and practical applications of the findings. Case studies or examples of current clinical trials could be beneficial. Can you provide examples of ongoing clinical trials or practical cases that have implemented these strategies, highlighting their practical impact?

Including diagrams or visual aids to illustrate the mechanisms discussed could make the information more accessible- the authors should consider that, visual aids can break down complex processes into more digestible parts

Author Response

Comment 1: The novel aspect of linking type-2 diabetes (T2D) with Alzheimer’s disease (AD) through microbiota-gut-brain communication pathways provides a fresh perspective that is not widely covered in the current literature. However,  there are some areas for improvement Including more discussions on conflicting results and different perspectives in the literature could provide a more balanced view. This would also help in highlighting the areas where further research is needed. Could you include a more detailed discussion of these results and the alternative viewpoints proposed in the literature in sections Gut microbiota-brain Axis or Diet and Transcription Factors in AD Progression sections

Response 1: Thank you for your comment. However, this review article provides a new perspective of gut-brain communication in terms of transcription factors as a commonly spoken language facilitates the interaction between gut and brain of obese diabetic patients who eventually develop cognitive impairment. That commonality is in terms of regulation of JAK3 expression both in gut and brain and hence maintaining the integrity of intestinal barrier and the phagocytic as well as migratory microglial functions, probably by regulation of microglial actin cytoskeleton remodeling, thereby alleviating the systemic chronic low-grade inflammation as well as Aβ deposition. Please find the modified title and  abstract.

comment 2: Including diagrams or visual aids to illustrate the mechanisms discussed could make the information more accessible- the authors should consider that, visual aids can break down complex processes into more digestible parts

Response 2: we have attached 2 graphs representative for the ideas of subtitle 4.2 "High fat diet-induced Gut Dysbiosis and Leaky Gut are involved in AD Pathogenesis" and of the title 5 "Perspective on HFD-dysregulated Transcription Factors Contribution in AD Progression" with figure legends.

Reviewer 3 Report

Comments and Suggestions for Authors

Dear Redactors,

Thank you very much for the opportunity to revise the article “Brain-gut and Microbiota-Gut-Brain Communication in Type-2 Diabetes linked Alzheimer’s Disease”.

Authors discussed modifiable factors such as Type-2 diabetes and diet and their implications in microbiota-gut-brain and brain-gut communication and cognitive functions of healthy brain and their dysfunction in Alzheimer’s Disease. Special emphasis has been given on elucidation of the mechanistic aspects of the impact of diet on gut-microbiota and the implications of some of the gut-microbial products in T2D and AD pathology.

The article is very interesting and well written. I have no specific comments. I just recommend to describe in more details effect of serotonin on brain physiology and AD development.

Thanks

Author Response

comment 1:  I just recommend to describe in more details effect of serotonin on brain physiology and AD development.

Response 1: Thank you for your reviewing and comment. However, this review article does not discuss the role of serotonin in the brain physiology and development. this is not relevant to the idea of our review.  . This review article provides a new perspective of gut-brain communication in terms of transcription factors as a commonly spoken language facilitates the interaction between gut and brain of obese diabetic patients who eventually develop cognitive impairment. That commonality is in terms of regulation of JAK3 expression both in gut and brain and hence maintaining the integrity of intestinal barrier and the phagocytic as well as migratory microglial functions, probably by regulation of microglial actin cytoskeleton remodeling, thereby alleviating the systemic chronic low-grade inflammation as well as Aβ deposition. Moreover, the characterization of those common transcription factors might represent a novel potential pharmacological target to impede AD progression.  

Round 2

Reviewer 1 Report

Comments and Suggestions for Authors

The revised manuscript has been improved significantly. However, there are still some concerns needed to be addressed.

1, the authors discussed the Gut-brain crosstalk in AD,  high fat diet and transcription factors in AD as well as the Janus kinase (JAK) in AD.  However, the linkage among gut-brain crosstalk, high fat diet and JAK kinase is weak. The authors should discuss more on the internal linkage among these events in AD. The authors should elaborate how high fat diet affect JAK kinase activity, which lead to dysregulation of gut-brain axis and AD onset. 

minor defects:

1, the absteact is too long. the authors should abbreviate it.

Author Response

comment 1: the authors discussed the Gut-brain crosstalk in AD,  high fat diet and transcription factors in AD as well as the Janus kinase (JAK) in AD.  However, the linkage among gut-brain crosstalk, high fat diet and JAK kinase is weak. The authors should discuss more on the internal linkage among these events in AD. The authors should elaborate how high fat diet affect JAK kinase activity, which lead to dysregulation of gut-brain axis and AD onset. 

Response 1: Please find the added sentences in red under subtitle 6.

Comment 2: the abstract is too long. the authors should abbreviate it.

Response 2: Sir/ Ma'am, I agree with you that the abstract is relatively long and I have worked to shorten it to adhere to the maximum allowed word numbers for the abstract but I believe shortening of the abstract more than that may distort the general meaning and idea of the review and this is what happened when it was firstly submitted and made the article less clear to be understood. Thank you for your consideration.
